# Analysis of Petiole Sap Nutrients Using Rapid and Standard Methods and Its Relation to Leaf Analysis of Fertilized *Malus domestica* cv. Gala

Mariana Mota [1], M. João Martins [2], Layanne Sprey [1], Anabela Maurício [3], Cristina Rosa [4], João Faria [4], Miguel B. Martins [1], Miguel L. de Sousa [5], Ricardo Santos [6], Rui M. de Sousa [5], Henrique Ribeiro [1] and Cristina M. Oliveira [1,*]

[1] LEAF—Linking Landscape, Environment, Agriculture and Food Research Center, Associated Laboratory TERRA, Instituto Superior de Agronomia, Universidade de Lisboa, Tapada da Ajuda, 1349-017 Lisboa, Portugal; mariana@isa.ulisboa.pt (M.M.); layannesprey14@gmail.com (L.S.); mmartins@isa.ulisboa.pt (M.B.M.); henriqueribe@isa.ulisboa.pt (H.R.)

[2] CEF—Centro de Estudos Florestais, Instituto Superior de Agronomia, Universidade de Lisboa, Tapada da Ajuda, 1349-017 Lisboa, Portugal; mjmartins@isa.ulisboa.pt

[3] FRUBAÇA-Cooperativa de Horto-Fruticultores, C.R.L. Lugar Acipreste Aptd. 12, 2460-471 Alcobaça, Portugal; anabela@copa.pt

[4] GRANFER-Produtores de Frutas, CRL. Rua Principal 167, 2510-772 Usseira, Portugal; cristina.rosa@granfer.pt (C.R.); joao.faria@granfer.pt (J.F.)

[5] INIAV, I.P., Polo Alcobaça, Estrada de Leiria, 2460-059 Alcobaça, Portugal; miguel.leao@iniav.pt (M.L.d.S.); rui.sousa@iniav.pt (R.M.d.S.)

[6] CAMPOTEC IN-Cons. e Transformação de Hortofrutícolas, SA. EN 9, 2560-393 Torres Vedras, Portugal; ricardosoaressantos@sapo.pt

* Correspondence: crismoniz@isa.ulisboa.pt

**Abstract:** Currently, fertilization decisions in apple orchards are based on soil and leaf analyses while the leaf material is sampled after the growing season, usually in June–July (90–110 days after full bloom). This approach is inefficient, as the information becomes available later than the growing season and is therefore only useful in supporting fertilization decisions for the next year, not the current one. To establish a method that provides useful information for fertilization decisions earlier in the growth cycle, our research focused on the assessment of the nutrient content of petiole sap using different methods, the standard method and the rapid method using a reflectometer. For this study, in 2021, four 'Gala' orchards were fertilized with different N–P–K levels. Macro and micronutrients were determined in leaves and petiole sap at 45 and 90–110 days after full bloom (DAFB) using standard laboratory methods and the reflectometer. When leaf analysis at 45 and 90–110 DAFB was compared with petiole sap analysis at the same time point, no significant correlations were found between the nutrient contents in leaf material and petiole sap, with the exception of calcium. However, positive results were obtained regarding the correlation between reflectometer determination and standard laboratory analyses. The regression analysis revealed high determination coefficients for N-NO$_3^-$ ($R^2 = 0.703$), K$^+$ ($R^2 = 0.705$), Ca$^{2+}$ ($R^2 = 0.715$), and Mg$^{2+}$ ($R^2 = 0.780$) between standard laboratory methods and the reflectometer. These results suggest that the reflectometer enables a real-time diagnostic tool for monitoring nutrient status throughout the growth cycle, particularly key nutrients related to fruit quality. The N–P–K fertilization strategies had no influence on the nutrient content of leaves or petiole sap. The nutrient content of both sample types varied depending on the orchard.

**Keywords:** nutrient analysis methods; fertilizer rate; ion-selective strips read using reflectometry; correlation

## 1. Introduction

Diagnostic methods concerning the nutritional status of fruit trees are crucial for sustainable production and the quality of the fruit. Methods that enable early detection

of deficiencies and toxicities, whose effects affect the production and quality of the fruit, and their timely corrections, are of great importance, especially in high-yielding apple orchards. In addition, these methods aid in avoiding fruit nutritional disorders. Mineral fertilization has effects on plant productivity, disease and pest control, soil microbial community composition, and environmental impacts [1]. On the other hand, organic alternatives to mineral N fertilizer such as the application of manure, compost, mulching, and cover crops have been promoted as ecologically sustainable orchard management practices [2]. Reducing nutrient inputs to orchard ecosystems without negative impacts on fruit quantity and quality is a priority.

The determination of apple nutritional status is based on leaf analysis at an advanced stage of the growth cycle, 90–110 days after full bloom (DAFB). The results of this analysis are very limited in terms of the possibility of interventions in the same year and only serve as a basis for a recommendation to fertilize the trees in the following year. An alternative that would enable real-time nutrient diagnosis could be petiole sap analysis. Esteves et al. (2021) [3] reviewed the methods for diagnosing nutrient petiole sap and their possible application in citrus crops, although most of the work on petiole sap nutrients has been carried out with vegetable and greenhouse crops, such as potato [4,5], pepper [6], and tomato [7–9] crops. As far as methods, are concerned, ion-selective electrode pocket meters such as LAQUAtwin (Horiba, Kyoto, Japan) are currently used to measure nutrient concentrations in plant sap and can serve as a basis for precise on-farm decisions [10]. For example, for tomatoes, the sap nutrient concentration can be recommended as the most sensitive nutritional diagnostic indicator in relation to the expected yield [11].

Regarding perennial fruit crops, the sap petiole approach was studied for grape vines [12,13]. Several companies offer comprehensive grape vine nutrient sap analysis; however, no scientific evidence has been published to support the extent of nutrient deficiencies or toxicity. When monitoring nutrient status, the results of the analysis should be compared to validated standard values to determine whether the values indicate a deficiency or toxicity of the nutrient. Although these standard values have been developed for leaf analysis [14], this work has not yet been done for grape sap analysis. Citrus sap analysis is an emerging technology, and although only one article has been published on the concentration of nutrients in the petiole sap of sweet oranges [15], it has recently attracted interest and was the subject of a project at the University of Florida. The objectives of this project were the development of a sampling strategy for sap testing, testing options, and initial guidelines for determining the mineral contents of grapefruit and mandarin crops [3,16].

In the past, analysis of vacuum-extracted xylem sap has been used to monitor mineral uptake status in apples [17–21]. However, these are laboratory methods and are not easily accessible to technicians and fruit growers. The petiole sap is obtained through cell disruption by pressing the petioles, and it consists of sap and cell contents. When analyzing petiole sap, several aspects must be taken into account, since its composition can vary due to various factors: the time of day, the organ sampled and its position in the plant, fertilization, climatic conditions, and other factors such as cultivar and rootstock, as well as the phenological stage, cultivation techniques, and irrigation systems [3,13,22,23]. Furthermore, as Nagarajah (1999) [12], who developed the strip method for determining $NO_3^-$ and $K^+$ in the petiole sap of sultana grape vines, pointed out, there is a need to develop validated standard values for different growth stages and other varieties. Cadahía (2008) [13] presented the sap nutrient concentrations of 'Red Globe', a table grape vine, during the growing cycle. In addition to method standardization and the development of standard values for perennial crops, the sensitivity of sap and leaf analysis to variations in fertilization regimes should also be investigated.

If a petiole sap test can be performed reliably and inexpensively, it would be possible to monitor the nutritional status of an orchard during the phenological phases of the annual cycle or the nutritional status of trees in response to cultural practices. A previous study [24] was carried out in 2019 with the aim of assessing the quality of the relationships

between the measured concentrations of ions in apple petiole sap, cv. Gala, using a rapid method RQflex® reflectometer, and their concentrations were measured using the reference laboratory methods (ICP-OES and VIS-UV spectrophotometry). The RQflex® kit consists of a reflectometer and corresponding ion-specific strips. The strips available from the supplier limit the number of analyzable ions. In apple petiole sap, high coefficients of determination ($R^2 > 0.80$) were achieved between the RQflex® and the reference laboratory methods when determining $N-NO_3^-$, $K^+$, $Mg^{2+}$, and $Ca^{2+}$ using this rapid method [24]. In the former study, Pearson correlation values between the RQflex® and laboratory methods were low for $PO_4^{3-}$ and $NH4^+$. This is because $PO_4^3$ occurs in the sap in inorganic and organic forms and the strips only quantify P in the phosphate form, while ICP-OES measures P in organic and inorganic forms. Furthermore, the proportion of organic and inorganic P sap forms is variable, and the ICP-OES P measurements indicate greater variation in the organic P forms [24]. For $N-NH4^+$ sap content, the quick test determination is particularly sensitive to the presence of $Fe^{2+}$ and $Fe^{3+}$, as indicated in the manufacturer's technical information, which could explain the poorer correlation between the $N-NH4^+$ determinations of VIS spectrophotometry and the rapid test strip method.

In this work, the research focus was on comparing leaf and sap analyses to assess the nutritional status of intensive Gala apple orchards fertilized with different fertilization regimes. In addition, we intended to determine whether reflectometer sap analysis is a good alternative tool for apple growers that enables a reliable and quick diagnosis of the nutritional status of apple trees.

## 2. Materials and Methods

### 2.1. Plant Material and Environmental Conditions, Experimental Layout, and Treatment Applications

This study was conducted in 2021 in four commercial 'Gala' orchards in the Alcobaça apple growing region. The trees were grafted onto M9 rootstock and were central leader trained. Table 1 shows the main characteristics of the orchards. The soils have a clay texture, and the pH value ($H_2O$) is between 7.3 and 8.3 (neutral to alkaline). These are soils with no or very low salinity (EC in water extract 1:2 < 0.44 mS cm$^{-1}$), and the percentage of organic matter (OM) is low (<2%). Regarding soil N–P–K macronutrients, (Table 1) the amount of nitrogen in both forms (as $N-NH_4$ and $N-NO_3$) is similar in all orchards and the levels of extractable K and P (ammonium lactate extraction [25]) are high. The climate of the Alcobaça region is a Csb climate (temperate climate with rainy winters and dry, mild summers) according to the Köppen–Geiger climate classification [26]. In the experimental region, the average annual temperature in 2021 was 14.7 °C, similar to the long-term average (1981–2010) of 15.0 °C. In that year, the total annual precipitation was 596.3 mm, which was low compared to the long-term period (1981–2010) average of 839.6 mm.

**Table 1.** Locations, planting date, 'Gala' clone, tree density, spacing, and soil N–P–K macronutrients.

| Orchard | Latitude Longitude | Planting Date | Clone | Area (ha) | Tree Density (Trees ha$^{-1}$) Spacing (m) | N-NH$_4$ | N-NO$_3$ mg kg$^{-1}$ | P$_2$O$_5$ | K$_2$O |
|---|---|---|---|---|---|---|---|---|---|
| A | 39°26′59.95″ N 9° 01′5.14″ W | 2016 | Schniga SchniCo(s) | 1.0 | 3759 3.80 × 0.70 | 10.9 | 3.3 | 552 | 437 |
| B | 39°30′55.01″ N 9°00′54.71″ W | 2016 | Schniga SchniCo(s) | 1.2 | 3565 3.30 × 0.85 | 9.5 | 12.7 | 958 | 393 |
| C | 39°28′30.48″ N 9°07′12.72″ W | 2015 | Brookfield | 4.5 | 2500 4.00 × 1.00 | 11.1 | 13.2 | 323 | 305 |
| D | 39°32′55.36″ N 8°57′22.52″ W | 2004 | Galaxy Selecta | 0.8 | 1851 4.50 × 1.20 | 9.7 | 15.7 | 391 | 213 |

The design of each apple orchard consisted of three randomized blocks (trial plots) per treatment. Each block consisted of a plot with 15 trees. The trees selected for sampling showed homogeneous vegetative growth and flowering intensity.

The treatments consisted of (i) standard fertilization according to the rules of integrated fruit production [27], (ii) double standard fertilization, and (iii) double standard fertilization in which part of the nutrients were in the form of organic materials (OMs). With standard fertilization, the amount of N, $P_2O_5$, and $K_2O$ applied depends on the expected production and the results of the foliar analysis; in addition, the amounts of N, $P_2O_5$, and $K_2O$ applied are corrected after considering the levels of these nutrients in the soil. In orchard A, soil $P_2O_5$ levels were high, so it was not technically advisable to further increase the amount of $P_2O_5$ applied.

In orchards A and C, organic fertilizer consisted of cow manure (5 t ha$^{-1}$), and in orchards B and D, the organic granular fertilizers were Organocad and Biofert (1.125 t ha$^{-1}$ and 1.5 t ha$^{-1}$, respectively). The N–P–K percentage was 3–2.4–12 for cow manure, 2.8–1.5–2.7 for Organocad, and 4.5–3–2 for Biofert. Table 2 shows the total amount of nutrients applied, with 50% applied to the soil as solid fertilizer in mid-March and 50% applied through fertigation from early May to mid-July, except for in orchard D where there was no fertigation. In orchard D, fertilization also took place during the growing season in April and June.

**Table 2.** Amount of nutrients (kg ha$^{-1}$) applied.

| Orchard | | N | $P_2O_5$ | $K_2O$ |
|---|---|---|---|---|
| | | (kg ha$^{-1}$) | | |
| | Standard | 63 | 50 | 73 |
| A | 2 × standard | 103 | 67 | 159 |
| | 2 × standard OM | 103 | 67 | 159 |
| | Standard | 49 | 19 | 75 |
| B | 2 × standard | 98 | 38 | 150 |
| | 2 × standard OM | 96 | 41 | 146 |
| | Standard | 45 | 23 | 64 |
| C | 2 × standard | 81 | 54 | 111 |
| | 2 × standard OM | 79 | 53 | 109 |
| | Standard | 51 | 22 | 73 |
| D | 2 × standard | 101 | 43 | 146 |
| | 2 × standard OM | 132 | 51 | 132 |

Calcium and boron were applied through foliar sprays according to commercial practice from fruit cell division to fruit cell enlargement. Magnesium (15–20 kg ha$^{-1}$) was also supplied. The trees were pruned, irrigated, and protected from pests and diseases according to local commercial practice.

Full bloom occurred on 8 April; 45 DAFB corresponds to the stage of fruit cell multiplication and 90–110 DAFB corresponds to the stage of fruit cell enlargement.

### 2.2. Leaf Sampling and Preparation

For sap analysis and leaf analysis, leaf samples were taken between 08:00 and 09:30 on sunny days. At 45 and 90–110 DAFB, for each analysis, 120 whole leaves were collected (leaf and petiole) (eight leaves per tree), corresponding to 120 leaves per replication and treatment. Leaves were collected when fully green, totally developed but not in senescence, at the same height in the tree or in a similar leaf position as much as possible to reduce variations. The leaves were transported to the Soil and Plant Chemistry Laboratory of the Instituto Superior de Agronomia, Lisbon, stored in thermal bags with ice, and processed the same day. For sap analysis, the petioles were separated from the leaf blades in the laboratory, cut into 0.5 cm pieces, and immediately stored in a −20 °C freezer for at least

24 h until the sap extraction and analysis. Petiole freezing pre-treatment facilitates the sap extraction process due to cell rupture [24]. For leaf analysis, whole leaves were placed in a forced air dryer at 65 °C for 48 h and ground.

### 2.3. Sap Extraction, RQflex® Reflectometer Procedure

In this study, the sap extraction method used was adapted from [12] and described in [24]. The petioles were crushed manually in a hydraulic press, then the sap was collected and analysed using a reflectometer (RQflex®, Merck, Darmstadt, Germany) and standard laboratory methods: (i) for $N-NO_3^-$, colorimetry was conducted in a continuous-flow analyser (Skalar SAN+, Skalar Analytical B.V., Breda, The Netherlands) using the sulfanilamide method [28]; (ii) for P, K, Ca, Mg, S, Fe, Mn, Zn, Cu, and B, inductively coupled plasma optical emission spectrometry was used (iCAP 7000 Series ICP Spectrometer, Thermo Fisher Scientific, Waltham, MA, USA).

In order to avoid sap oxidation measurements with the reflectometer test strips, the preparation for the laboratory analysis was carried out immediately. Due to the high sap concentration, dilution with deionized water was required for the $Ca^{2+}$, $Mg^{2+}$, and $K^+$ determination according to Table 3 so that the concentration of the diluted sample was in the middle of the analytical range. The $N-NO_3^-$ measurement with RQflex® was carried out with undiluted fresh sap (Table 3). For the standard methods, petiole sap was diluted in nitric acid for all elements except $N-NO_3^-$, where the dilution was in potassium chloride (Table 3). The RQflex® strips were dipped in the sap and developed color within a standard time, and the color intensity was measured with a reflectometer. The amount of light reflected by the strip was measured and converted into a concentration through pre-calibration, making the method quantitative. Although strips are available for P and $N-NH_4^+$, previous results did not show good correlation with reference methods for these two ions [24] and these nutrients were therefore not measured with the reflectometer.

**Table 3.** Dilution factors for 'Gala' apple sap for measurement with the reflectometer and reference methods.

| | Dilution Factors ($\times$) | | | |
|---|---|---|---|---|
| Reflectometry dilution with deionized water | $N-NO_3^-$ | $Ca^{2+}$ | $Mg^{2+}$ | $K^+$ |
| 45 DAFB | 1 | 5 | 5 | 10 |
| 90–110 DAFB | 1 | 10 | 10 | 20 |
| Standard methods ICP-OES | - | | 20 in $HNO_3$ (5%) | |
| VIS spectrophotometry | 20 in KCl (2 M) | | - | |

DAFB (days after full bloom); ICP-OES (inductively coupled plasma optical emission spectrometry); VIS (visible).

Calibration with standard solutions was previously performed by comparing the test strips results with ICP-OES and VIS spectrophotometry according to Almeida et al. (2020) [24]. The macronutrient concentration is expressed in mg $L^{-1}$ except for $K^+$, which is expressed in g $L^{-1}$.

### 2.4. Leaf Mineral Analysis

Leaf samples were previously dried at 65 °C (until constant weight) and ground in a stainless-steel mill (Cullatti model TYP MFC, Cullatti, Steinen, Switzerland) with a 1.0 mm sieve. For N determination, 0.25 g of dried material was digested in 4 mL of $H_2SO_4$ with selenium as a catalyzer following the Kjeldahl method [29] and quantified using segmented flow analysis through colorimetry in a continuous-flow analyser (Skalar SAN+ Systems, Skalar Analytical B.V., Breda, The Nederland) according to the Berthelot method [28]. For P, K, Ca, Mg, S, Fe, Mn, Zn, Cu, and B determination, 0.30 g of dried material was digested in a mix of HCl (12 mL) and HNO3 (4 mL) at 105 °C in a block digestion system (Digipress MS, SCP Science, Baie-d'Urfé, QC, Canada) following an adapted version of the CSN EN 13650.

After digestion, the elements were quantified using inductively coupled plasma optical emission spectrometry (iCAP 7000 Series ICP Spectrometer, Thermo Fisher Scientific, USA). The macronutrient concentration is expressed as % DW and the micronutrient concentration is expressed as mg kg$^{-1}$ DW.

Figure 1 summarizes the materials and methods used.

- Four orchards
- Three fertilization strategies: standard, 2 × standard and 2 × standard OM
- Leaves collected at 45 and 90-110 DAFB

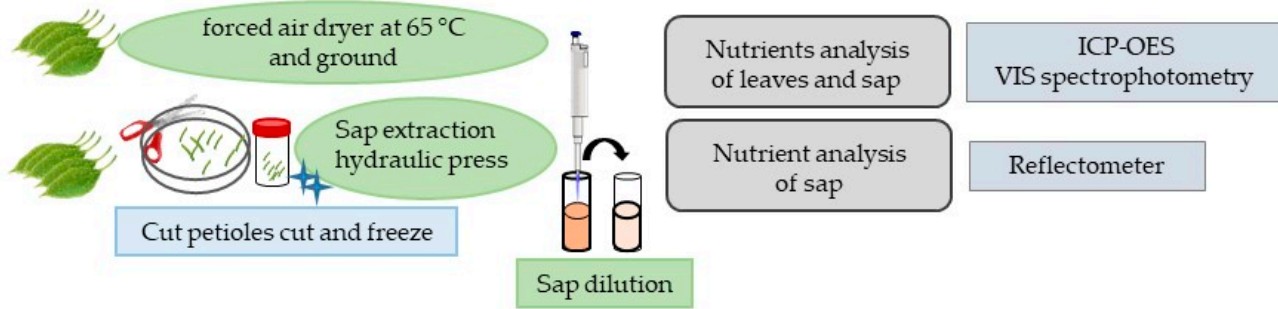

**Figure 1.** Summary of the materials and methods used.

### 2.5. Statistical Analysis

All data were stored in R objects (https://CRAN.R-project.org (accessed on 3 August 2023)) and all statistical analyses and graphs were produced in R. The Pearson correlation coefficient was computed and tested for its significance using the cor and cor.test functions. Means were compared using ANOVA models that were fitted with aov function. Whenever the factors had principal or interaction significant effects (F-tests with *p*-values less than 0.05), the Tukey's test was applied in order to detect the pairs of means that were significantly different. Tukey's tests were performed with the HSD.test function from the agricolae package (https://CRAN.Rproject.org/package=agricolae (accessed on 3 August 2023)).

### 3. Results and Discussions

#### 3.1. Leaf Analysis and Petiole Sap Analysis through Reference Methods

3.1.1. Macro- and Micronutrient Contents in Leaves and in Petiole Sap

Leaf macro- and micronutrient contents obtained through reference methods at 45 DAFB and 90–110 DAFB are shown in Table 4. Some elements such as Ca, Mg, and Fe increased in the leaves between these time points. Between these phenological stages, there was a decrease in leaf N, K, Cu, Zn, and B contents. With the exception of P, whose values are slightly above the reference value, all values in both data sets were within the reference value interval. These intervals or these sufficiency ranges refer to the optimal element concentration range below which deficiency occurs and above which toxicity or imbalances occur. Although there were variations in the concentrations of the various nutrients (slight increases and decreases, depending on the nutrient), the variations are not substantial, and the values are comparable within the range of standard values. This suggests that leaf mineral concentration values in these two developmental stages were not substantially different, with the exception of Ca, where the concentration was higher in the latter sampling stage (due to foliar sprays). Previous work with 'Gala' apples [30] that used averages of four-year data to compare 45 DAFB leaves and 90–110 DAFB leaves had similar results. According to Nachtigall and Dechen (2006) [31], the apple leaf nutrient concentrations throughout the vegetative cycle presented relative stability after the 10th week after full bloom (70 DAFB), also indicating that leaf sampling

for nutritional diagnosis might be anticipated for 30 days in relation to the standard assessment at 90–110 DAFB.

**Table 4.** Macro- (%) and micronutrient (mg kg$^{-1}$) analyses in leaves at 45 and 90–110 DAFB, the mean ± SD, and the leaf concentration standards for 'Gala' at the 90–110 DAFB stage [27]. The values reported are for dry weights.

| Nutrient | 45 DAFB | 90–110 DAFB | Leaf Concentration Standards at 90–110 DAFB |
|---|---|---|---|
| N (%) | 2.57 ± 0.196 | 2.52 ± 0.262 | 2.50–3.00 |
| P (%) | 0.206 ± 0.0212 | 0.183 ± 0.0165 | 0.14–0.18 |
| K (%) | 1.91 ± 0.218 | 1.60 ± 0.147 | 1.30–2.00 |
| Ca (%) | 0.957 ± 0.182 | 1.44 ± 0.173 | 0.90–1.6 |
| Mg (%) | 0.246 ± 0.0373 | 0.311 ± 0.0353 | 0.20–0.30 |
| S (%) | 0.208 ± 0.0172 | 0.210 ± 0.0310 | 0.22–0.30 |
| Fe (mg kg$^{-1}$) | 72.7 ± 24.2 | 110 ± 31.8 | >45 |
| Cu (mg kg$^{-1}$) | 12.4 ± 3.90 | 11.5 ± 1.58 | 10–50 |
| Zn (mg kg$^{-1}$) | 46.7 ± 20.5 | 36.5 ± 6.51 | 10–100 |
| Mn (mg kg$^{-1}$) | 183 ± 84.7 | 200 ± 68.6 | 25–200 |
| B (mg kg$^{-1}$) | 33.2 ± 10.4 | 28.0 ± 4.89 | <50 |

Sap macro- and micronutrient contents obtained using reference methods at 45 DAFB and 90–110 DAFB are shown in Table 5. In general, there was an increase in macronutrients between these time points. The same was observed for micronutrients. Although our work is based on data from 2021 and the data shown in previous work [24] is from 2019, the values are not very different.

**Table 5.** Macro- and micronutrient analyses (mg L$^{-1}$) of sap at 45 and 90–110 DAFB and the mean ± SD of 36 values.

| Nutrient | 45 DAFB | 90–110 DAFB |
|---|---|---|
| N-NO$_3$$^-$ | 9.84 ± 8.25 | 16.9 ± 7.68 |
| P | 228 ± 32.7 | 282 ± 56.6 |
| K | 6868 ± 565 | 8411 ± 604 |
| Ca | 1234 ± 142 | 2080 ± 212 |
| Mg | 356 ± 50.4 | 595 ± 65.3 |
| S | 85.6 ± 19.5 | 164 ± 54.0 |
| Fe | 2.89 ± 0.94 | 3.69 ± 1.13 |
| Cu | 1.08 ± 0.370 | 3.50 ± 4.47 |
| Zn | 11.7 ± 6.40 | 10.4 ± 3.00 |
| Mn | 15.8 ± 8.47 | 26.8 ± 12.3 |
| B | 131 ± 14.2 | 175 ± 28.3 |

### 3.1.2. Effect of Fertilization on Nutrients in Leaves

Considering the elements measured in the leaves, Table 6 presents the comparison of nutrient contents between orchards and fertilization strategies for each time period. In general, the fertilization strategy had no effect on either date (45 and 90–110 DAFB) as the averages in each row share some letters. The concentrations of the elements varied depending on the orchard. For N, there were differences between orchards at 90 DAFB, for P there were differences at both time points, and for K there were differences only at 45 DAFB.

### 3.1.3. Effect of Fertilization on Nutrients in Petiole Sap

In general, different strategies did not influence nutrient contents in sap (Table 7). Other work on citrus [15] claimed that sap analysis was very sensitive to variations in the fertilization system; in particular, N and K showed an increase with increasing nutrient supply, although treatments in this experiment ranged from 0% (no nutrients)

to 200% of the standard fertilizer dose corresponding to expected production. In our work, the range of fertilizer quantities was not as wide. However, there were significant differences between orchards. The differences between orchards for P and K were more noticeable at 90–110 DAFB than at 45 DAFB. The amount of nutrients applied at 45 DAFB had no immediate influence on the content of nutrients in the sap; for instance, the N content was different in orchards A and D, with orchard A having the highest values of $N\text{-}NO_3^-$ ($p < 0.001$) in petiole sap and orchard D having the lowest value ($p < 0.001$). At 90–110 DAFB, all orchards were N fertilized, and orchard A showed the highest value of $N\text{-}NO_3^-$ ($p < 0.001$) compared with the other orchards. The highest value of P and K in petiole sap was found in orchard C ($p < 0.001$) at 90–110 DAFB. Apparently, there is no relationship between the analysis of the petiole sap and the date or amount of fertilizer application. In 2021, the trees from orchards A and B were 5 years old, the trees from orchard C were 6 years old, and the trees from orchard D were 17 years old. This means that there is generally no direct response of nutrient levels in petiole sap to the timing and dose of fertilizer applied, regardless of tree's age.

**Table 6.** Averages of three replicates of the content of each nutrient (% dry weight) in leaves for each orchard and fertilization strategy. Different letters mean that the means are significantly different according to Tukey's post hoc tests.

| | 45 DAFB | | | 90–110 DAFB | | |
|---|---|---|---|---|---|---|
| | **Standard** | **2 × Standard** | **2 × Standard OM** | **Standard** | **2 × Standard** | **2 × Standard OM** |
| Orchard | | **N** | | | **N** | |
| A | 2.61 [ab] | 2.63 [ab] | 2.57 [ab] | 2.60 [abc] | 2.49 [abcd] | 2.54 [abc] |
| B | 2.64 [ab] | 2.40 [ab] | 2.45 [ab] | 2.35 [cd] | 2.41 [bcd] | 2.41 [bcd] |
| C | 2.34 [b] | 2.63 [ab] | 2.40 [ab] | 2.07 [d] | 2.34 [cd] | 2.54 [abc] |
| D | 2.75 [ab] | 2.84 [a] | 2.57 [ab] | 2.72 [abc] | 2.90 [a] | 2.87 [ab] |
| | | **P** | | | **P** | |
| A | 0.191 [d] | 0.191 [d] | 0.191 [d] | 0.176 [cde] | 0.168 [de] | 0.171 [de] |
| B | 0.218 [bc] | 0.201 [cd] | 0.199 [cd] | 0.176 [cde] | 0.169 [de] | 0.163 [e] |
| C | 0.197 [cd] | 0.185 [d] | 0.188 [d] | 0.188 [bcd] | 0.179 [bcde] | 0.186 [bcd] |
| D | 0.252 [a] | 0.225 [b] | 0.230 [ab] | 0.200 [ab] | 0.198 [abc] | 0.216 [a] |
| | | **K** | | | **K** | |
| A | 1.94 [bcde] | 1.88 [cdef] | 2.13 [ab] | 1.69 [a] | 1.73 [a] | 1.61 [a] |
| B | 1.71 [efg] | 1.98 [abcd] | 1.83 [defg] | 1.63 [a] | 1.74 [a] | 1.57 [a] |
| C | 1.68 [fg] | 1.59 [g] | 1.72 [efg] | 1.67 [a] | 1.53 [a] | 1.38 [a] |
| D | 2.22 [a] | 2.09 [abc] | 2.18 [ab] | 1.61 [a] | 1.56 [a] | 1.54 [a] |

For each element and period (45 or 90–100 DAFB) the letters refer to all cells, with each cell corresponding to a pair (orchard, strategy).

### 3.1.4. Nutrient Correlations in Leaf and Petiole Sap Analyses

Leaf analysis measures everything in a given sample, both inorganic and organic compounds. In addition to nutrient absorption through the roots, calcium and boron sprays were also carried out. In leaves, the correlation between the concentration of few elements (Figure 2) was quite high and positive; this was the case for Ca and Mg (r = +0.870), while the correlation was negative for Mg and K, for example (r = −0.738). This linear positive correlation between Ca and Mg found in angiosperms could potentially be related to cell wall chemistry [32]. Mg and K being highly negatively correlated may be explained by the antagonisms between these cations in root absorption and transport within plants [33]. Boron content was correlated, albeit moderately, with Mg (r = −0.639), Mn (r = −0.625), and Ca (r = −0.595). Boron was applied through foliar sprays because it plays an important role in cell walls and in reproductive tissues.

**Table 7.** Averages of three replicates of the content of each nutrient (mg L$^{-1}$) in petiole sap for each orchard and fertilization strategy. Different letters mean that the means are statistically different according to Tukey's post hoc tests.

| | 45 DAFB | | | 90–110 DAFB | | |
|---|---|---|---|---|---|---|
| | **Standard** | **2 × Standard** | **2 × Standard OM** | **Standard** | **2 × Standard** | **2 × Standard OM** |
| Orchard | | **N-NO$_3$$^-$** | | | **N-NO$_3$$^-$** | |
| A | 20.2 [a] | 20.6 [a] | 20.0 [a] | 24.8 [b] | 30.1 [ab] | 32.3 [a] |
| B | 2.02 [c] | 2.08 [c] | 1.00 [c] | 13.4 [c] | 11.2 [c] | 12.7 [c] |
| C | 11.3 [b] | 11.6 [b] | 12.2 [b] | 14.9 [c] | 13.6 [c] | 14.1 [c] |
| D | 1.19 [c] | 0.267 [c] | 0.149 [c] | 10.2 [c] | 10.7 [c] | 14.6 [c] |
| | | **P** | | | **P** | |
| A | 198 [bc] | 181 [c] | 223 [abc] | 233 [cd] | 247 [cd] | 289 [bc] |
| B | 206 [abc] | 213 [abc] | 207 [abc] | 340 [ab] | 276 [c] | 260 [cd] |
| C | 2709 [a] | 239 [abc] | 258 [ab] | 344 [ab] | 346 [ab] | 372 [a] |
| D | 253 [ab] | 233 [abc] | 251 [ab] | 229 [cd] | 200 [d] | 250 [cd] |
| | | **K** | | | **K** | |
| A | 6898 [bc] | 6821 [c] | 7087 [abc] | 8197 [cd] | 8190 [cd] | 8115 [bc] |
| B | 5363 [abc] | 7113 [abc] | 6698 [abc] | 8370 [ab] | 8634 [c] | 8309 [cd] |
| C | 6764 [a] | 7440 [abc] | 7297 [ab] | 8839 [ab] | 9361 [ab] | 9215 [a] |
| D | 7125 [abc] | 6631 [ab] | 7183 [ab] | 7596 [cd] | 8390 [cd] | 7714 [d] |

For each element and period (45 or 90–100 DAFB) the letters refer to all cells, with each cell corresponding to a pair (orchard, strategy).

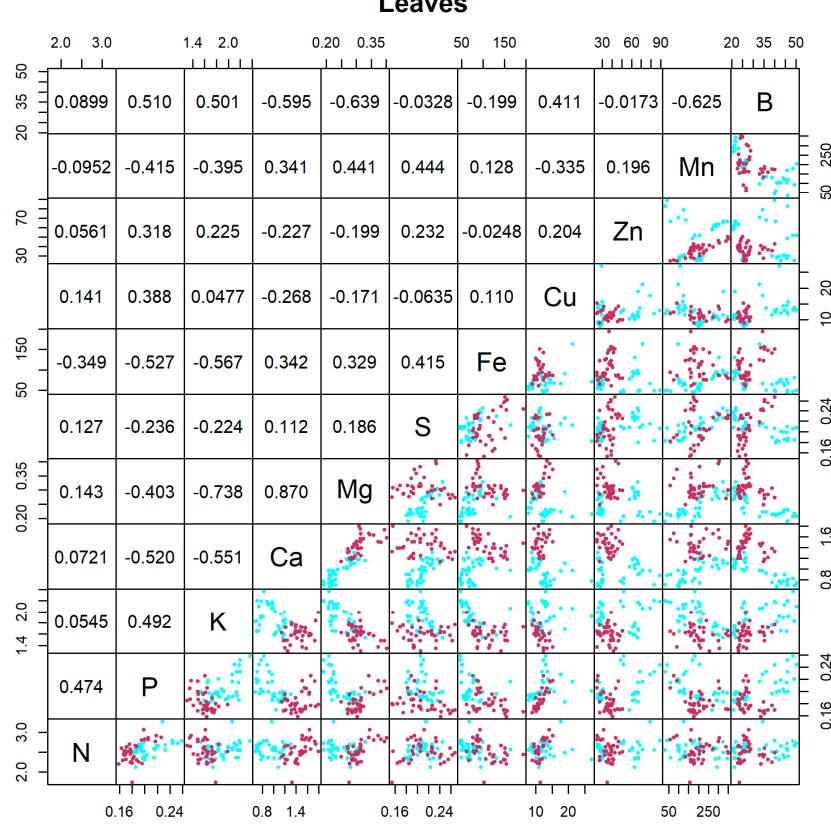

**Figure 2.** Matrix of scatterplots between pairs of elements in leaves (n = 72). The scatterplot in line i and column j (j > i) indicates the content of element i plotted against content of element j. Cyan is used for 45 DAFB and maroon is used for 90 DAFB. The upper panel displays the corresponding Pearson correlation coefficients, where line i and column j (i > j) contain the correlation between the contents of elements i and j. Correlations greater than 0.195 in absolute value are significantly different from zero (*p* < 0.05).

Sap analysis is a real-time picture of current nutrient status for tree growth and development and reflects inorganic compounds. Looking at sap analysis using standard methods (Figure 3), the correlation values between the concentrations of Ca and Mg (r = +0.962) and Ca and K (r = +0.833) are very high. Some correlations which were negative for leaf analysis are positive and high, as in the case of K and Mg (r = +0.812), possibly due to Mg fertilization (15 kg ha$^{-1}$), as this fertilization is necessary to alleviate K-induced Mg deficiency. These results suggest that the effect of fertilizer application may be reflected in different rhythms in leaves and petiole sap. The boron concentration was negatively correlated with the concentrations of all elements with moderate correlation values, such as Ca (r = −0.656), Mg (r = −0.625), N (r = −0.637), and K (r = −0.631). The concentrations of Cu and S were highly correlated (r = +0.846), possibly due to copper sulfate sprays to control diseases.

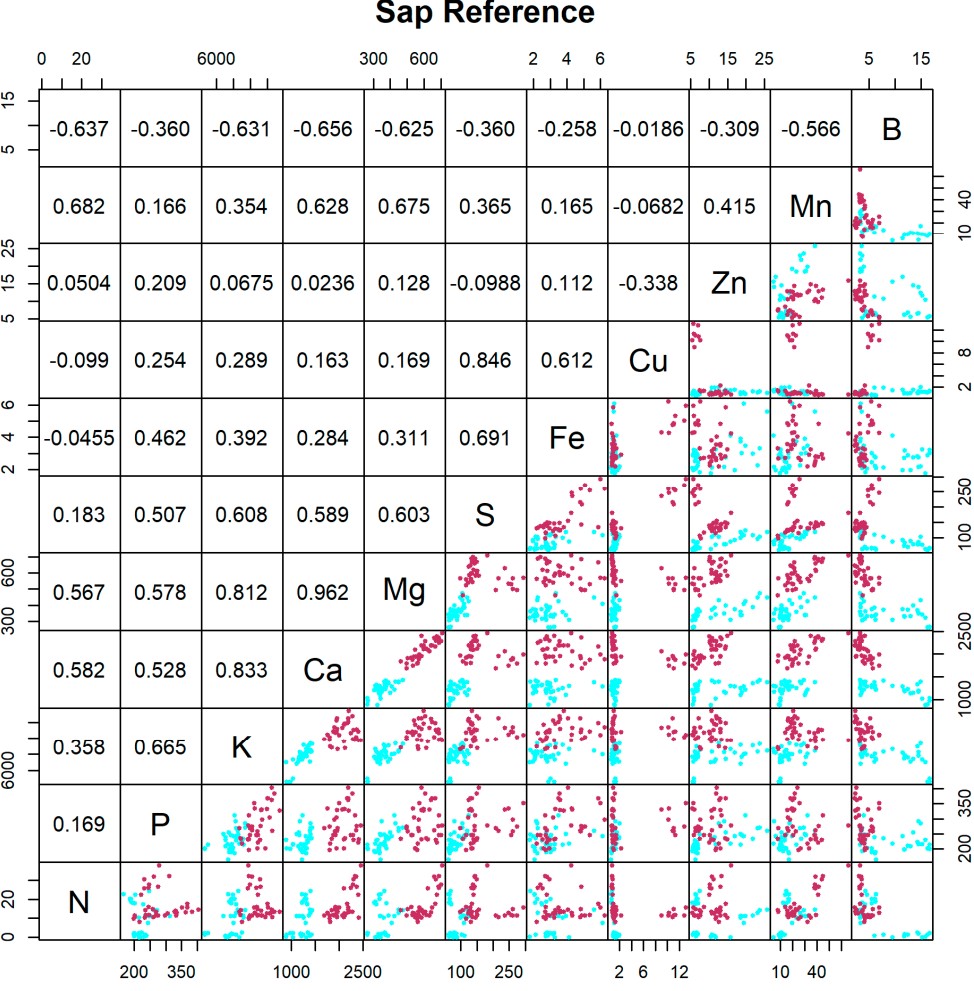

**Figure 3.** Matrix of scatterplots between pairs of elements in sap obtained using the reference method (n = 72). The scatterplot in line i and column j (j > i) indicates the content of element i plotted against the content of element j. N is N-NO$_3^-$. Cyan is used for 45 DAFB and maroon is used for 90 DAFB. The upper panel displays the corresponding Pearson correlation coefficients, where line i and column j (i > j) contain the correlation between the contents of elements i and j. Correlations greater than 0.195 in absolute value are significantly different from zero (*p* < 0.05).

3.1.5. Relation between Macro- and Micronutrients in Leaf and Petiole Sap Analyses

The relationship between sap and conventional tissue nutrient analysis is variable in crops and depends on the developmental growth stage. The relationship between leaf analysis and sap analysis is shown in Figure 4. When merging data from 45 and 90–110 DAFB, the Pearson correlation coefficient (r) and probability values from linear

regression between leaf and sap elements obtained using reference methods are high for Ca (r = 0.809, *p* < 0.001) and moderate for Mg (r = 0.644, *p* < 0.001), B (r = −0.640, *p* < 0.001), and Cu (r = 0.633, *p* < 0.001). In apple cv. Gala, the petiole sap Ca concentration correlated well with the Ca leaf tissue concentration [24].

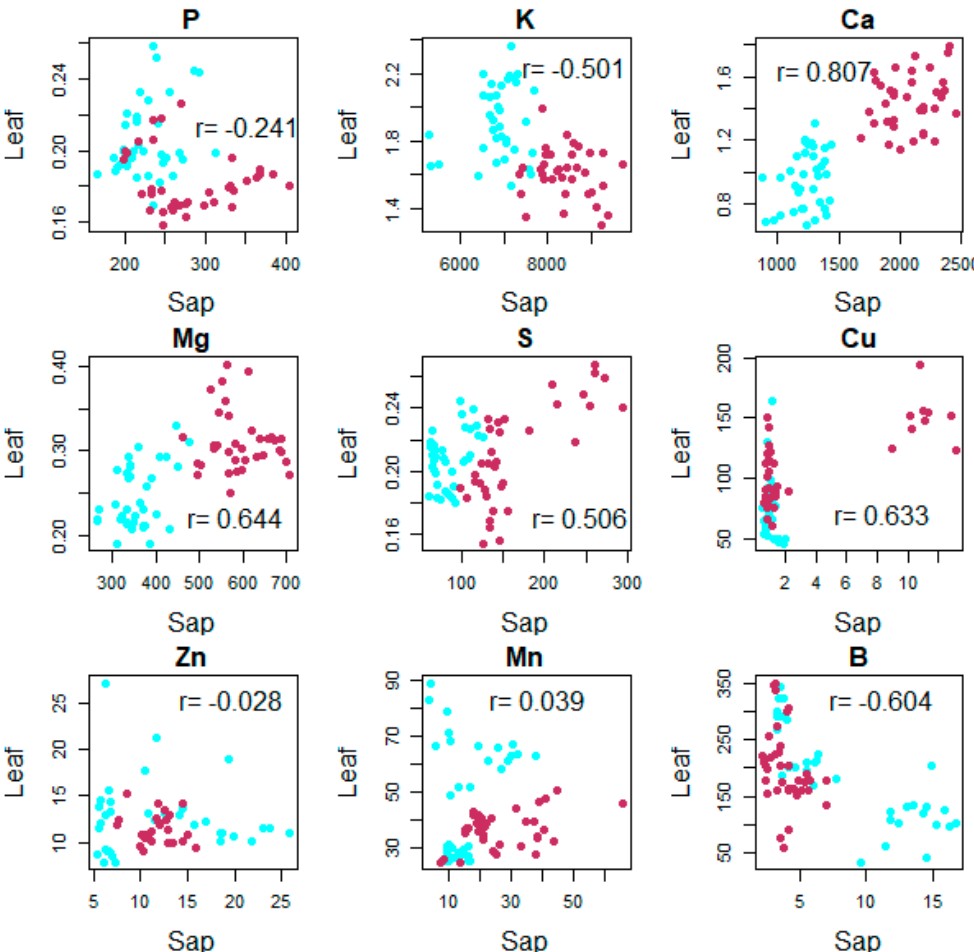

**Figure 4.** Relationship between the nutrient contents in leaves and in sap for both periods (cyan is used for 45 DAFB and maroon is used for 90 DAFB) (n = 72). Correlations greater than 0.195 in absolute value are significantly different from zero (*p* < 0.05).

### 3.2. Relationship between the Petiole Sap Content of Each Element Determined with the Reference Method and with the Reflectometer

To verify the accuracy between the data obtained with the test strips and with the standard methods, Figure 5 shows the linear regression between the data obtained with the two methods by combining the data from 45 and 90–110 DAFB. The regression analysis revealed high determination coefficients for N-NO$_3^-$ ($R^2$ = 0.703), K$^+$ ($R^2$ = 0.705), Ca$^{2+}$ ($R^2$ = 0.715), and Mg$^{2+}$ ($R^2$ = 0.780) between the reflectometer and laboratory methods.

In a previous work concerning the Gala apple variety [24], the regression analysis also resulted in high determination coefficients between the RQflex® and laboratory methods for these ions, reinforcing the concept of using this quick method to monitor fertilization, particularly the amount of nitrogen applied, in order to reduce leaching, which poses a major environmental risk. On the other hand, these ions are very important minerals related to fruit quality [34]. This tool enables real-time in situ analysis, allowing farmers to easily assess nutritional status throughout the growing cycle. This provides useful information, particularly in high-density orchards where fertilizer rates tend to be high.

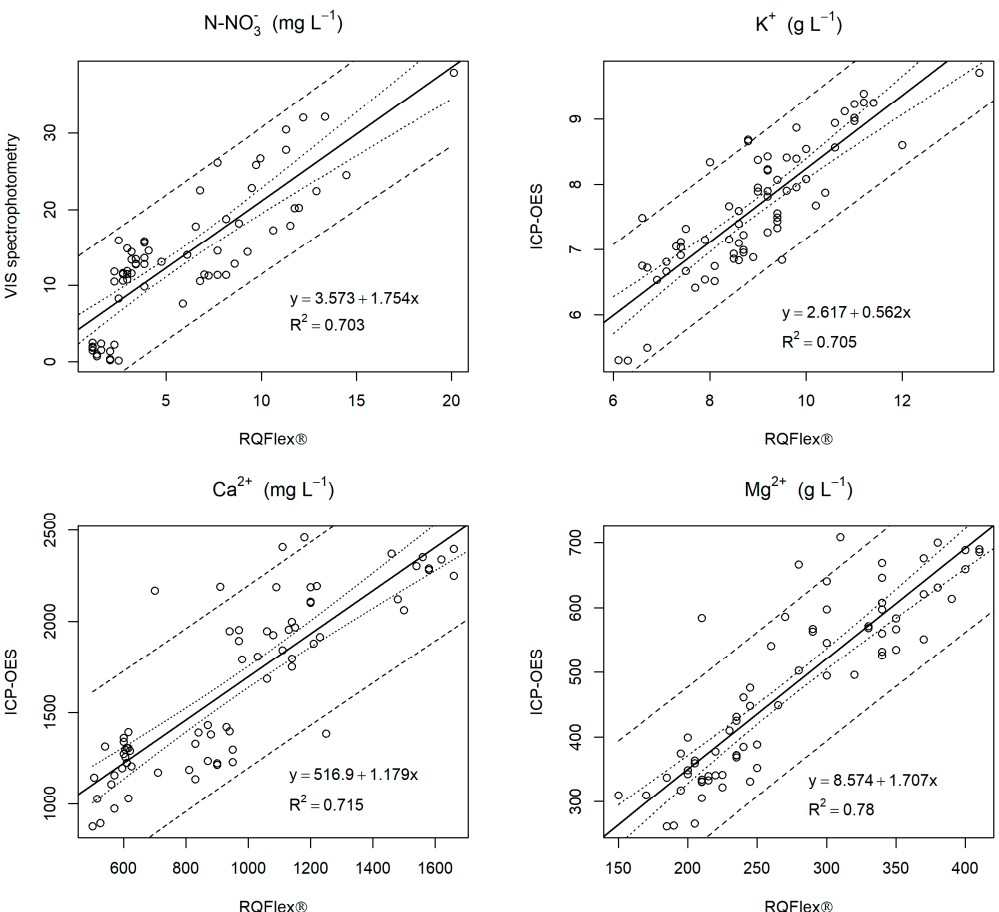

**Figure 5.** Relationships between petiole sap contents of each element obtained using the reference method (vertical axis) and using the reflectometer (horizontal axis RQflex$^{®}$ test strips) for $N - NO_3^-$, $K^+$, $Ca^{2+}$, and $Mg^{2+}$. Each plot shows the fitted line, the confidence bars, and the prediction bars at 95%. The equation of the fitted line and its precision are also shown. For each element, 72 observations were used.

## 4. Conclusions

This is the second study to analyse apple petiole sap in the frame of nutritional diagnostics. This study was conducted in different apple orchards and under different fertilization regimes in order to evaluate the potential of the putative method. With the exception of calcium, no significant correlations were found between the nutrients in the leaf material and petiole sap when comparing leaf and petiole sap analysis at 45 and 90–110 DAFB. This aspect results from the fact that the nutrients contained in leaf tissue reflect plant accumulation rather than indicating the real-time concentration available for plant development and highlights the need to establish validated reference values for petiole nutrient contents obtained from apple orchards under different growing conditions for use in petiole sap analysis in fertilization studies and diagnosis. The N–P–K fertilization strategies had no influence on the nutrient contents of the leaves or petiole sap. The nutrient contents of both sample types varied depending on the orchard.

This is the first time that a rapid method for determining nutrient contents in petiole sap has be successfully used in a large sample size. Regression analysis revealed high determination coefficients ($R^2 > 0.7$) for $N\text{-}NO_3{}^-$, $K^+$, $Ca^{2+}$, and $Mg^{2+}$ between the standard laboratory methods and the reflectometer method. This shows that the field estimations of petiole sap nutrient contents based on the reflectometer measurements are reliable and provide real-time and in situ information on nutrient statuses in orchards concerning the most important nutrients related to fruit quality. This information can be used to directly assess the evolution of nutrient ranges throughout the growth cycle. To utilize the

reflectometer, fruit growers must consider standardizing the protocol, namely, collecting the leaf samples early in the morning, freezing the petioles, and extracting the petiole sap using a manual hydraulic press or equivalent device, and then following the manufacturer's instructions. This equipment can complement soil and foliar analysis and provide site-specific information to support optimal fertilization practices in modern high-density apple orchards. The reflectometer method should be applied to other fruit species to determine its universality.

**Author Contributions:** Conceptualization: C.M.O., M.M. and M.J.M.; Data Curation: M.J.M., M.M. and C.M.O.; Funding acquisition: C.M.O., A.M., C.R., M.L.d.S. and R.S.; Investigation: L.S., A.M., C.R., J.F., M.L.d.S., R.S. and M.B.M.; Methodology: C.M.O., H.R., A.M., C.R., M.L.d.S., R.S. and R.M.d.S.; Project administration: C.M.O.; Resources: H.R. and R.M.d.S.; Supervision: C.M.O.; Writing—Original Draft: C.M.O., M.M. and M.J.M.; Writing—critical review and editing, C.M.O. All authors have read and agreed to the published version of the manuscript.

**Funding:** This research was supported with national funding by Programa de Desenvolvimento Rural—PDR 2020 within the MACFERTIQUAL PDR2020-101-031590 project and by FCT—Fundação para a Ciência e a Tecnologia, I.P., under the project UIDB/04129/2020 of LEAF-Linking Landscape, Environment, Agriculture and Food, Research Unit.

**Data Availability Statement:** Data are contained within the article.

**Conflicts of Interest:** Anabela Maurício was employed by the company FRUBAÇA, Cristina Rosa and João Faria from GRANFER, and Ricardo Santos from CAMPOTEC IN declare no conflict of interest. The remaining authors declare that the research was conducted in the absence of any commercial or financial relationships.

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
