# Peer review of "Analysis of Petiole Sap Nutrients Using Rapid and Standard Methods and Its Relation to Leaf Analysis of Fertilized Malus domestica cv. Gala"

_horticulturae, doi:10.3390/horticulturae10010036_

Round 1
Reviewer 1 Report
Comments and Suggestions for Authors
The manuscript examines the possibilities of rapid analysis of nutrients in petiole sap sample of apple using the reflectometer. The presented method is interesting and provides apple growers with a tool that allows accurate and rapid diagnosis of the nutritional status of apple trees and enables more precise fertilization to reduce environmental impact.
I suggest “Graphical abstract” from Line 47 to Line 49 be moved to section "Materials and Method"
Table 2. Amount of nutrients (kg ha-1) applied. Line 150 it needs to be formatted
I recommend the authors to apply the method to other fruit species to determine its universality.
Author Response
Thank you for revising our manuscript. Attached you will find the letter answering your questions.

Reviewer 2 Report
Comments and Suggestions for Authors
Manuscript ID: horticulturae-2738279
Title: Analysis of petiole sap nutrients by rapid and standard methods and relation to leaf analysis of fertilized Malus domestica cv. Gala
The main topic of the article is the evaluation of alternative methodologies for the early diagnosis of the nutritional status of apple trees aimed at the possibility of applying corrective measures in time for a high and eco-sustainable production yield.
Although the topic may be quite interesting, it must be said immediately that the article is poorly developed, and the many different arguments touched upon are not well connected to each other. The many topics covered, such as sustainable productivity, quality of fruit, fertilisation levels, comparison of nutrient analysis in leaves and petiole sap with traditional and rapid methods (reflectometer), are developed in little depth and not well related to each other. Many terms such as standard, sensitivity, reference value, etc. do not seem to be appropriate in the context to which they refer.The Abstract and Introduction sections need to be revised and rewritten. Due to the numerous topics touched upon, the true purpose of the article is not clear, nor are the expectations and novelties related to the study conducted. There is a poor discussion and the conclusions are rather lacking in originality and not always supported by the results
Below are some points where action is needed:
In general, the abstract needs to be rewritten better: many statements are disconnected from each other and the discourse is not well developed and linear. Many terms such as "accuracy", "reference values", do not seem to be appropriate.
Line 39 what is "...the most important nutrient related to fruit quality".
The Introduction is also poorly written and needs to be revised and expanded. In particular, there is insufficient information on which individual parameters, or a combination of them, are fundamental for defining the nutritional status of the plant in real time so that useful corrections can be made in relation to specific diseases, nor are examples of other such crops included.
Furthermore:
Lines 56-57: the concept of eco-sustainability must be expanded
Lines 58-61: this statment is unclear.
Lines 62-65: pooly structured sentence.
Line 67: " no standards" ?
Line 83: " ...to develop standards" ?
Lines 84-87: what do you meant by " sensitivity of sap and leaf analysis to..." ?
Lines 88-91: very cofusing sentence.
Line 91: "As a preliminary work"? why preliminary?
Lines 96-97: "The strips available from the supplier limit the ions tested" why tested, perhaps "analyzable".
Line 97-98 : "In the present study, the sap extraction method was adapted from [6] and described in [17]"; this sentence should be entered in Methods and not in Introduction.
Lines 101-103: the sentence is unclear.
Line 107: "in sap analysis" ? why in? perhaps "of".
Lines 110-117: very unclear; but what is the real purpose of this work? what new things are expected in relation to what is already known about the various topics discussed?
2. Materials and Method
Line 127: "the amount of nitrogen forms is similar in all orchards and the levels of extractable K and P (ammonium lactate extraction [18]) are high." But what are these values? it would be appropriate to include them in Table 1.
Line 142: what is standard fertilisation? the reference 20 is a 'Normas Tecnicas' in portuguese and cannot be found at the address entered. However, it is good that the authors give a more accurate description of the fertilisation plan. Furthermore, the values entered in table 2 are either wrong or not easily understood for what little is included in the text.
Lines 215-218: it is advisable to add a brief description of the instrumentation, methods and chemicals used.
Lines 245: the authors should explain why they have included this table which shows for each nutrient analyzed the mean value on orchard and fertilization strategy, mean value which therefore corresponds to the average of the 12 numbers given in table 6.
Better explicit what "the Reference Value"
Lines 235-236: the standard deviations for many parameters are very high (>10-30%); it is therefore wrong to give significance to the differences the authors point out between the two samplings. Lines 238-240: this statement contrasts with the previous one.
Line 242: clarify the sentence.
Line 248: as with table 4, the authors should explain why, for each nutrient analysed, the average value on orchard and fertilisation strategy was presented in table 5, values which then correspond to the average of the 12 numbers in each cell, shown in table 7.
Line 248: Tab 5 shows the results of sap analysis carried out with the same traditional methods used for leaves; for nitrogen, the authors used the Kjeldhal method, which allows the quantification of total nitrogen bound to proteins. In table 5, data related to nitrate nitrogen (N-NO3) are included: how were they obtained? Furthermore, how do the data obtained with standard laboratory methods and those determined with the Reflectometer for this parameter compare in 3.2 in view of the above? Lines 252-255: what does this sentence have to do with this context? what data is there to support this statement?
Lines 262-266: The comment to Table 6 is unclear. As the authors report that the fertilization strategy had no effect for each date (45 and 90-110 DAFB) on the element concentration and that the orchard influenced the element concentration, the values of the average orchard effect should be presented. However, if a significant interaction Fertilisation Strategy X Orchard was observed (which is not stated by the authors), the presentation of the data in Table 6 would be appropriate.
Line 321: also in table 7, the nitrogen content is incorrectly stated as N-NO3 even though it is determined by the Kjeldhal method.
Line 307: is unclear " since there was no application of N at 45 DAFB in orchard A and D".
Lines 381-393: the authors continue to erroneously relate the values of nitrogen obtained by standard laboratory and reflectometer analysis to each other; the former is referred to as N-Kjeldhal (protein- bound nitrogen) and the latter as nitrate nitrogen.
Line 399-401: the authors stated that “This tool, combined with soil and foliar analysis, can provide real-time in situ analysis and allow farmers to determine the best fertilization practices, especially in high-density orchards where fertilizer rates tend to be high”, This statement does not appear to be supported by the results, as the authors reported that “The fertilization strategy had no influence on the mineral content of the leaves and sap”. How can this tool enable farmers to determine the best fertilisation practices if “the fertilisation strategy did not affect the mineral content of leaves and sap”?
The conclusions need to be expanded and in them the novelties derived from the experimental work described should be better stated, especially in comparison to what has already been written in the recent literature (reference 17).
Author Response

(The authors gave the same response as above.)

Reviewer 3 Report
Comments and Suggestions for Authors
Review
The manuscript submitted for review entitled "Analysis of petiole sap nutrients by rapid and standard methods and relation to leaf analysis of fertilized Malus domestica cv. Gala” is interesting, but its main drawback is the fact that the results only come from an experiment conducted for one year. This fact reduces scientific credibility resulting from repetitions and variability in nature in subsequent growing seasons. Having results based on a single year's experiment might limit the depth and robustness of the findings, particularly in agricultural studies where variations across growing seasons can significantly impact outcomes.
The manuscript has a standard layout, in my opinion the authors clearly formulated the purpose of the research,
- the work lacks detailed information on the physical and chemical properties of the soils on which the experiment was carried out. The authors included only general information, it is not known whether the differences in properties were statistically significant - please complete.
- Mineral analysis of leaves was done according to an adapted version of the European 215 standard EN 13650- please describe briefly method,
- please round the analysis results, e.g. in table 4, to three significant places (all tables)
- Line 252 -values are not very different, in my opinion you should write whether the differences are statistically significant or statistically insignificant,
In my opinion, figures (1-3) are not very legible
- In my opinion, conclusions should be reworded. In their conclusions, the authors should include clear statements that they have reached as a result of their research. Encouraging the authors to refine the conclusions to explicitly present the core findings and their broader implications would enhance the clarity and impact of the manuscript, allowing readers to grasp the significance of the research more effectively.
I included the remaining comments in the attached PDF file

Author Response

(The authors gave the same response as above.)

Round 2
Reviewer 2 Report
Comments and Suggestions for Authors
Manuscript ID: horticulturae-2738279
Title: Analysis of petiole sap nutrients by rapid and standard methods and relation to leaf analysis of fertilized Malus domestica cv. Gala
Major
The authors replay to my comments and advice are not such as to change my mind about this work.
It is the case that same of the changes I suggested have been made, but the major observations especially regarding the experimental design description, the motivation and originality of the results and conclusions, has not substantially changed. In essence, the authors do not clearly articulate what the main and especially innovative findings are that make the publication of their research significant.
Minor
Lines 24-26: DAFB must be written in full
Lines 55-56: the sentence is unclear
Line 77 and 95: for clarity, "values" should be added after "standard"
in Line 79: add a reference after "... have been developed for leaf analysis"
Line 152-154: even though the authors have replaced the reference previously mentioned but not found on the web with another one that can be found, I remain of the opinion that it is appropriate to give more guidance on the fertilisation plan. The values entered in table 2 are unclear: for example, how come the amount of P added in A (67 kg per ha) relative to 2 x standard fertilisation is not double that indicated for standard? What are the timescales for soil fertilisation and fertigation?
Line 328: why "....and both had no N applied at 45 DAFB". What is the timing of fertilisation and fertigation?
Line 355-359: this sentence is not significant within the context in which it is inserted
Line 370-372: it is not clear why” Some correlation which were negative for 370
leaf analysis are positive a... is possible with Mg fertilisation (15 kg ha-1)…"
Line 394-397: the sentence is out of context
In Fig 5 what is the difference between Nx and N (this applies to all parameters)? You must specify in the label. Also wrong to put N (N-NO3 is correct)
Line 422-424: In response to my observation, the authors comment that "Vegetable crops respond quickly to fertilization practices; therefore, monitoring the
sap nutrient content is a well-established tool to provide information about current
nutritional status. In fruit crops, the response of trees to fertilization regimes is less clear
due to the perennial structure. However, the diagnosis of nutritional status is important
and forms the basis for fertilization". This is precisely why more clarification is needed on the importance of 'this tool' and what 'useful information' is. It is too general in the text.
Line 426-447: the conclusions have been rewritten but in essence have not improved: the authors should try to enhance the results obtained in a way that emphasizes the novelty of the research carried out. As they themselves reiterated, the real problem is to create standard protocols that will allow the information from nutrient analysis in petiole sap in this type of crop to be used to the best advantage, as is already practiced for vegetable crops. Nothing new is added in the conclusions in this regard. The use then of test kits to carry out some analyses in real time, can certainly help, but the validity and usefulness of this goes hand in hand with solving the problem said first.
Author Response
Dear reviewer, thank you for our work in revising the manuscript. Attached you will find the new version, the answer to your question, and the new Figure 5. Best regards, the authors.

Reviewer 3 Report
Comments and Suggestions for Authors
Review
The authors revised the manuscript according to my suggestions. I recommend the submitted manuscript for further stages of editorial processing.
Author Response
Dear Reviewer
Thank you for considering the manuscript for editing
best regards, the authors